# FROM BASIS TO BASIS: GAUSSIAN PARTICLE REPRESENTATION FOR INTERPRETABLE PDE OPERATORS

## ABSTRACT

Learning PDE dynamics for fluids increasingly relies on neural operators and Transformer-based models, yet these approaches often lack interpretability and struggle with localized, high-frequency structures while incurring quadratic cost in spatial samples. We propose to represent fields with a *Gaussian basis*, where learned atoms carry explicit geometry (centers, anisotropic scales, weights) and form a compact, mesh-agnostic, directly visualizable state. Building on this representation, we introduce a *Gaussian Particle Operator* that acts *in modal space*: learned *Gaussian modal windows* perform a Petrov–Galerkin measurement, a *PG Gaussian Attention* effects global cross-scale coupling. This basis-to-basis design is resolution-agnostic and achieves near-linear complexity in $N$ for fixed modal budget, supporting irregular geometries and seamless 2D→3D extension. On standard PDE benchmarks and real datasets, our method attains state-of-the-art–competitive accuracy while providing intrinsic interpretability.

## 1 INTRODUCTION

Fluid-governed PDEs (Wazwaz, 2002; Gurtin, 1982) underpin critical real-world systems, from numerical weather prediction and climate reanalysis to ocean circulation and engineering aerodynamics (McKeown et al., 2020; Shlesinger et al., 1987). Classical solvers (finite element/volume and spectral methods) (Wazwaz, 2002; Johnson, 2012; Klaasen & Troy, 1984) deliver high fidelity but face persistent challenges: strongly multi-scale dynamics, mesh dependence and complex geometries, stiffness in time integration, and high computational cost for long rollouts. Neural operators (Li et al., 2021; Kovachki et al., 2023; Lu et al., 2021) emerged as data-driven maps between function spaces, enabling resolution-agnostic surrogates; more recently, Transformer-based operators (Cao, 2021; Hao et al., 2023) leverage attention to capture long-range interactions and achieve strong empirical performance on diverse PDE tasks. However, these models still suffer from two key limitations: *(i) poor interpretability*—latent features and attention weights are rarely tied to physically meaningful modes; and *(ii) localization/frequency bias*—global self-attention tends to favor low-rank, low-frequency correlations, making sharp fronts, vortical filaments, and other high-frequency structures harder to model, while naïvely scaling attention over $N$ spatial samples incurs $\mathcal{O}(N^2)$ cost (Li et al., 2025).

We advocate representing fluid fields with a *Gaussian (particle) basis* rather than a fixed grid, hand-picked spectra (Gupta et al., 2021; Li et al., 2024), or a monolithic implicit network (Serrano et al., 2023; 2024). Gaussian atoms carry *explicit geometry*—centers and (anisotropic) scales—which align naturally with coherent flow structures (vortices, filaments, fronts), afford multiscale locality, and are directly *visualizable* and *differentiable*. This basis is meshagnostic and compact, supports irregular boundaries, and extends seamlessly from 2D to 3D (Buhmann, 2000; Park & Sandberg, 1991). While prior neural representations often rely on global Fourier features, wavelets, or blackbox INRs, *learning a particleized Gaussian basis as the primary state of the field* has been scarcely explored and offers a clearer bridge to physical intuition. Concretely, a field is approximated by weighted Gaussians with $\mu_i$ (centers), $\sigma_i$ (scales, possibly anisotropic), and $w_i$ (mixture weights) learned from data; evaluating these atoms at query locations yields a compact coefficient vector that serves as the field's interpretable latent state (basis).

We present an interpretable, resolution-agnostic neural operator that learns a *Gaussian particle* basis for fields and couples it with a *Petrov–Galerkin Gaussian Attention* layer, enabling basis-to-basis

modeling with near-linear complexity and strong accuracy on 2D/3D and irregular domains. Our contributions are summarized as follows:

(1) **Gaussian Particle Representation.** An encoder learns per-site Gaussians $(\mu, \sigma, w)$; evaluating at arbitrary queries yields an interpretable, visualizable basis $Z$ that is mesh-agnostic and extends seamlessly to 3D.

(2) **PG Gaussian Attention.** Learned *Gaussian modal windows* perform PG-style measurement $(N \to G)$, a $G \times G$ attention implements the modal kernel (global coupling), and the result is scattered back $(G \to N)$, yielding a principled and interpretable operator.

(3) **Efficiency & Scalability.** With a small modal budget $G \ll N$, spatial transfers scale $\mathcal{O}(N)$ and modal attention is independent of $N$, delivering near-linear growth with resolution and supporting multi-step operator stacking.

(4) **Empirical validation.** Across standard PDE benchmarks and real datasets (including ERA5 and 3D/irregular domains), our approach attains state-of-the-art–competitive accuracy while providing *intrinsic interpretability* (particle and modal diagnostics), yielding improved spectral fidelity and rollout stability—demonstrating a favorable accuracy–interpretability trade-off.

## 2 RELATED WORK

**Neural operators.** Classical neural operator methods aim to learn mappings between function spaces directly from data, typically by parameterizing a resolution-agnostic kernel or lifting to a latent space and learning integral transforms. Representative approaches include the *Fourier Neural Operator* (FNO), which performs global convolution via spectral multipliers to approximate operator kernels in Fourier space (Li et al., 2021; Kovachki et al., 2023), and *DeepONet*, which decomposes an operator into branch/trunk networks to separately encode input functions and query coordinates (Lu et al., 2021). Variants extend these ideas with multiresolution bases (Li et al., 2020b; Gupta et al., 2021; He et al., 2024; Li et al., 2024), graph or kernelized message passing (Li et al., 2025), and learned Green's functions (Li et al., 2020a).

These models are *purely data driven*: while many designs are physics-inspired, their internal representations are typically opaque. In particular, the learned latent bases and mixing weights are not tied to interpretable physical primitives, which limits diagnostic insight and the ability to attribute predictions to physically meaningful components.

**Transformer-based methods.** A recent line of work adopts Transformers to parameterize neural operators, replacing hand-crafted kernel parameterizations with data-driven attention. Examples include *Galerkin Transformers*, which align attention with variational forms (Cao, 2021), *GNOT* (Hao et al., 2023) that leverage attention for long-range coupling, *Transolver* (Wu et al., 2024), which introduces slice-based attention for efficient global mixing, and operator networks that stack attention with physics-informed objectives (Xiao et al., 2024). Empirically, with sufficiently large training corpora and careful scaling, attention-based operators often match or surpass traditional neural operators in expressive power and generalization to out-of-distribution forcings and grids.

However, these gains come with two well-known limitations. **(i) Lack of interpretability:** standard attention weights are not anchored to physically interpretable trial/test functions, making it difficult to ascribe predictions to identifiable modes or localized mechanisms. **(ii) Frequency bias:** global self-attention tends to emphasize low-rank, global correlations (low-frequency structure), while recovering sharp, localized, or high-frequency phenomena often requires architectural add-ons or extensive data augmentation. As a result, pure Transformer operators provide limited physical attribution and may under-represent fine-scale features without additional inductive biases.

## 3 METHODOLOGY

### 3.1 GAUSSIAN BASIS REPRESENTATION

**From physics field to Gaussian field and basis.** We represent a spatial field $a : \Omega \subset \mathbb{R}^d \to \mathbb{R}^{d_a}$ by a set of *Gaussian particles* placed at each sample location $\mathbf{x}_j$. Each particle is parameterized by

a center $\mu_{j,i} \in \mathbb{R}^d$, axis-aligned scale $\sigma_{j,i} \in \mathbb{R}^d$, and mixture weight $w_{j,i} \in [0,1]$ with $\sum_{i=1}^G w_{j,i} = 1$. The associated (unnormalized) kernel is

$$G(\tilde{\mathbf{x}}; \mu_{j,i}, \sigma_{j,i}) = \exp\left(-\tfrac{1}{2}\big\|(\tilde{\mathbf{x}} - \mu_{j,i})/\sigma_{j,i}\big\|_2^2\right). \tag{1}$$

Evaluating these kernels at query $\tilde{\mathbf{x}}_j$ yields the *Gaussian basis* coefficients

$$z_{j,i} = w_{j,i}\, G(\tilde{\mathbf{x}}_j; \mu_{j,i}, \sigma_{j,i}), \qquad \mathbf{z}_j = [z_{j,1}, \ldots, z_{j,G}]^\top \in \mathbb{R}^G. \tag{2}$$

Physically, the Gaussian field acts as a mollified, locally supported expansion of $a(\cdot)$; the coefficients in Eq.(2) can be viewed as localized averages of $a$ under data-adaptive windows $(\mu, \sigma)$, while $w$ distributes mass among overlapping particles. This basis is resolution-agnostic and naturally extends to irregular geometries.

**Encoder.** Given samples $\{(\mathbf{x}_j, a_j)\}_{j=1}^N$ with $a_j \in \mathbb{R}^{d_a}$, an encoder $E_\theta$ first lifts features through a shared MLP $\phi_{\mathrm{in}} : \mathbb{R}^{d_a} \to \mathbb{R}^h$ and then branches into three heads for particle parameters:

$$\phi_{\mathrm{in}}(a_j) = \mathrm{ReLU}\big(\mathrm{Linear}(d_a, h)\, a_j\big), \tag{3}$$

$$f_\mu(\phi_{\mathrm{in}}(a_j)) = \mathrm{Linear}(h, h) \xrightarrow{\mathrm{ReLU}} \mathrm{Linear}(h, Gd), \tag{4}$$

$$f_\sigma(\phi_{\mathrm{in}}(a_j)) = \mathrm{Linear}(h, h) \xrightarrow{\mathrm{ReLU}} \mathrm{Linear}(h, Gd) \xrightarrow{\mathrm{Softplus}}, \tag{5}$$

$$f_w(\phi_{\mathrm{in}}(a_j)) = \mathrm{Linear}(h, h) \xrightarrow{\mathrm{ReLU}} \mathrm{Linear}(h, G) \xrightarrow{\mathrm{Softmax}}, \tag{6}$$

reshaped as $\mu_{j,i}, \sigma_{j,i} \in \mathbb{R}^d$ and $w_{j,i} \in [0,1]$. The Softplus ensures $\sigma_{j,i} > 0$; the Softmax normalizes mixture weights $\sum_i w_{j,i} = 1$.

**Gaussian basis evaluation.** With $(\mu, \sigma, w)$ predicted by $E_\theta$, the weighted Gaussian evaluation Eq.(2) produces the per-site latent vector $\mathbf{z}_j$. *Physically*, $\mu$ encodes particle locations, $\sigma$ controls receptive-field sizes (anisotropy along axes), and $w$ balances overlapping contributions. *Computationally*, the map $(\mu, \sigma, w, \tilde{\mathbf{x}}) \mapsto \mathbf{z}$ is local and embarrassingly parallel.

**Decoder.** A lightweight MLP head $f_\phi^{\mathrm{dec}} : \mathbb{R}^G \to \mathbb{R}^{c_{\mathrm{out}}}$ regresses from $\mathbf{z}_j$ to the field value at the query:

$$\hat{a}(\tilde{\mathbf{x}}_j) = f_\phi^{\mathrm{dec}}(\mathbf{z}_j). \tag{7}$$

In practice, we use a two-layer perceptron with ReLU.

**Gaussian particle regularization.** Because the constraints act on the *particle parameters* produced by the encoder, we impose them at the Gaussian-field level (conceptually tied to $E_\theta$ but applied after parameter prediction):

$$\text{(center alignment)} \quad \mathcal{L}_\mu = \frac{1}{N} \sum_{j=1}^N \Big\| \sum_{i=1}^G w_{j,i} \mu_{j,i} - \mathbf{x}_j \Big\|_2^2, \tag{8}$$

$$\text{(scale range)} \quad \mathcal{L}_\sigma = \frac{1}{NGd} \sum_{j,i,\ell} [\max(0, \sigma_{j,i,\ell} - \sigma_{\max}) + \max(0, \sigma_{\min} - \sigma_{j,i,\ell})], \tag{9}$$

which promote spatial interpretability (centers near coordinates), avoid degenerate particles, and discourage overly peaky mixtures.

**Overview pipeline.** Eqs.(2–7) define the Gaussian-field training pipeline, and the complete forward/backward diagram is in Figure 1. We minimize the reconstruction loss together with $\mathcal{L}_\mu, \mathcal{L}_\sigma$.

**Approximation capacity of the Gaussian basis.** We record a standard density result:

**Lemma 3.1** (Density of Gaussian mixtures). *Let $\Omega \subset \mathbb{R}^d$ be compact. Finite mixtures of (possibly anisotropic) Gaussians are dense in $C(\Omega)$ and in $L^r(\Omega)$ for $1 \leq r < \infty$. Consequently, for any continuous scalar field $v$ and $\varepsilon > 0$, there exist $G$ and parameters $\{(\mu_i, \sigma_i, w_i)\}_{i=1}^G$ such that*

$$\Big\| v(\cdot) - \sum_{i=1}^G w_i \exp\left(-\tfrac{1}{2} \|(\cdot - \mu_i)/\sigma_i\|_2^2\right) \Big\|_\infty < \varepsilon.$$

*Vector-valued fields admit componentwise approximation.* (Proof in Appx.A.1)

Figure 1: **Overview of the Gaussian Basis Field framework.** Given the physics field $a(x)$, an encoder $E_\theta$ produces $G$ Gaussian components per spatial location (mean $\mu$, scale $\sigma$, and mixture weight $w$). These define a Gaussian field that is evaluated at queries to form the basis $z$, which is decoded by a fixed decoder $f_\phi^{\text{dec}}$ to reconstruct the output field.

## 3.2 PETROV–GALERKIN GAUSSIAN ATTENTION

Section 3.1 established *local*, per-point Gaussian bases: at each location $j$, $G$ weighted coefficients $\mathbf{z}_j \in \mathbb{R}^G$ are derived from particles $(\mu, \sigma, w)$. To learn a resolution-agnostic operator on this basis, we adopt the Petrov–Galerkin (PG) perspective—in which a field is approximated in a trial space and residuals are enforced to be orthogonal to a (possibly different) test space—and move from the spatial grid to the *operator (modal) space* (Franca et al., 2006; Brooks & Hughes, 1982). Concretely, learned Gaussian modal windows first *measure* the field by aggregating information across locations, a global *mode coupling* operates on the $G$ Gaussian components, and the result is then *scattered back* to locations. This PG-guided pipeline yields a basis-to-basis operator that is both computationally efficient ($G \ll N$) and interpretable.

### 3.2.1 FROM PETROV–GALERKIN PROJECTION TO A GAUSSIAN-BASIS OPERATOR

In Petrov–Galerkin (PG), a field is expanded in a *trial* space and tested by a (possibly different) *test* space. Here, the local Gaussian particles serve as trial functions, while *Gaussian modal windows* act as discrete test functions that aggregate information from locations to global modes.

**Trial functions (Gaussian basis).** Let the unnormalized Gaussian particle (anchored at location $j$, component $i$) be

$$\phi_{j,i}(\mathbf{x}) = \exp\left( -\tfrac{1}{2}\left\| (\mathbf{x} - \mu_{j,i})/\sigma_{j,i} \right\|_2^2 \right), \qquad \Sigma_{j,i} = \text{diag}(\sigma_{j,i}^2). \tag{10}$$

With weighted evaluations (Sec. 3.1), each site provides $\mathbf{z}_j = [z_{j,1}, \ldots, z_{j,G}]^\top \in \mathbb{R}^G$, where $z_{j,i} = w_{j,i}\, \phi_{j,i}(\tilde{\mathbf{x}}_j)$.

**Test functions (Gaussian modal windows).** Define $G$ learned windows $\{\psi_g\}_{g=1}^G$ that softly select content across locations:

$$\psi_g(\mathbf{x}) \approx \sum_{j=1}^N \tilde{p}_{j,g}\, \delta(\mathbf{x} - \tilde{\mathbf{x}}_j), \qquad \tilde{p}_{j,g} = \frac{p_{j,g}}{\sum_{j'} p_{j',g}}, \tag{11}$$

where $p_{j,g} \geq 0$ and $\sum_g p_{j,g} = 1$ implement a soft assignment from locations to modes. Using a linear projection of coefficients $\mathbf{s}_j = \mathbf{z}_j W_z \in \mathbb{R}^D$, the PG *measurement* (test of the trial field) yields modal tokens

$$t_g = \frac{\sum_{j=1}^N p_{j,g}\, \mathbf{s}_j}{\sum_{j=1}^N p_{j,g}} \in \mathbb{R}^D, \qquad T = [t_1, \ldots, t_G] \in \mathbb{R}^{G \times D}. \tag{12}$$

**Modal coupling and scatter.** Let $\kappa : \{1, \ldots, G\}^2 \to \mathbb{R}^{D \times D}$ be a (learned) coupling kernel over modes. PG updates the modal state and scatters it back:

$$U_g = \sum_{g'=1}^G \kappa(g, g')\, t_{g'} \in \mathbb{R}^D, \qquad \widetilde{\mathbf{z}}_j = \left( \sum_{g=1}^G p_{j,g}\, U_g \right) W_{\text{out}} \in \mathbb{R}^G. \tag{13}$$

Stacking sites gives $\widetilde{Z} \in \mathbb{R}^{N \times G}$. Algebraically,

$$\widetilde{Z} \approx A\,\mathcal{K}\,A^\top\,(Z\,W_z)\,W_{\text{out}}, \qquad A[j,g] = p_{j,g}, \ \mathcal{K}[g,g'] \text{ encodes modal coupling.} \tag{14}$$

Thus PG supplies the *structure*: test (measure) $\to$ couple $\to$ scatter.

### 3.2.2 ATTENTION AS A PARAMETERIZATION OF THE PG OPERATOR

We now instantiate Eq.( 14) with a multi-head attention layer that is global in *modal* space and local in the $N \leftrightarrow G$ transfers. Let $Z \in \mathbb{R}^{N \times G}$ and particle parameters $(\mu, \sigma, w) \in \mathbb{R}^{N \times G \times d} \times \mathbb{R}^{N \times G \times d} \times \mathbb{R}^{N \times G}$.

**Learned Gaussian modal windows.** For head $h$, form a per-site descriptor $\xi_j = [\mathbf{z}_j, \mathbf{w}_j, \mu_j, \sigma_j] \in \mathbb{R}^{G(2d+2)}$ and project to $h_j^{(h)} \in \mathbb{R}^D$. A softmax over modes produces windows

$$p_{j,g}^{(h)} = \text{softmax}_g\big(W_p^{(h)} h_j^{(h)}\big), \tag{15}$$

which instantiate the PG test functions (Eq.( 11)) in discrete form. $W_p^{(h)} \in \mathbb{R}^{G \times D}$ is the (head-specific) linear projection that maps the local embedding $h_j^{(h)}$ at location $j$ to mode logits over the $G$ Gaussian modes.

**PG measurement $N \to G$.** Project coefficients $\mathbf{s}_j^{(h)} = \mathbf{z}_j W_z^{(h)}$ and compute tokens

$$t_g^{(h)} = \frac{\sum_j p_{j,g}^{(h)}\,\mathbf{s}_j^{(h)}}{\sum_j p_{j,g}^{(h)}} \in \mathbb{R}^D, \qquad T^{(h)} = [t_1^h, \ldots, t_G^h] \in \mathbb{R}^{G \times D}, \tag{16}$$

which matches the PG measurement in Eq.( 12).

**Global modal coupling ($G \times G$ attention).** Scaled dot-product attention parameterizes the kernel $\mathcal{K}$:

$$Q^{(h)} = T^{(h)} W_Q^{(h)}, \quad K^{(h)} = T^{(h)} W_K^{(h)}, \quad V^{(h)} = T^{(h)} W_V^{(h)}, \tag{17}$$

$$\alpha^{(h)} = \text{softmax}\left(\frac{Q^{(h)} K^{(h)\top}}{\sqrt{D}}\right), \tag{18}$$

$$\widetilde{T}^{(h)} = \alpha^{(h)} V^{(h)} \ \in \ \mathbb{R}^{G \times D}. \tag{19}$$

Here $\alpha^{(h)}(g, g')$ plays the role of a data-driven modal coupling $\kappa(g, g')$.

**Scatter $G \to N$ and readout.** Using the same windows, scatter the coupled modes back and read out to $G$ coefficients:

$$y_j^{(h)} = \sum_g p_{j,g}^{(h)} U_g^{(h)} \in \mathbb{R}^D, \qquad \widetilde{\mathbf{z}}_j = \left(\big\|_{h=1}^H y_j^{(h)}\right) W_{\text{out}} \in \mathbb{R}^G, \qquad \widetilde{Z} = [\widetilde{\mathbf{z}}_1^\top; \ldots; \widetilde{\mathbf{z}}_N^\top]. \tag{20}$$

To stabilize training and preserve the per-site total mass (row-wise $\ell_1$ sum), we first take a convex residual update with a mixing coefficient $\lambda \in [0, 1]$ and then renormalize each row:

$$\widehat{Z} = (1 - \lambda)\,Z \ + \ \lambda\,\widetilde{Z}, \tag{21}$$

$$Z'_{j,:} = \frac{\sum_{g=1}^G Z_{j,g}}{\sum_{g=1}^G \widehat{Z}_{j,g} + \varepsilon}\,\widehat{Z}_{j,:}, \qquad j = 1, \ldots, N, \tag{22}$$

where $\varepsilon > 0$ avoids division by zero. Eq.( 21) provides a conservative blend between the old and updated coefficients, while Eq.( 22) rescales each site's coefficients so that $\sum_g Z'_{j,g} = \sum_g Z_{j,g}$.

*Complexity.* With a fixed modal budget $G$ and head width $H \cdot D$, the two $N \leftrightarrow G$ transfers scale as $\mathcal{O}(B\,H\,N\,G\,D)$—*linear in $N$*—while modal attention is $\mathcal{O}(B\,H\,G^2 D)$ and *independent of $N$*. Hence when $N$ grows sharply (e.g., dense global atmospheric grids, from 2D to 3D meshes), compute/memory remain near-linear in resolution, in contrast to spatial self-attention's $\mathcal{O}(B\,H\,N^2 D)$.

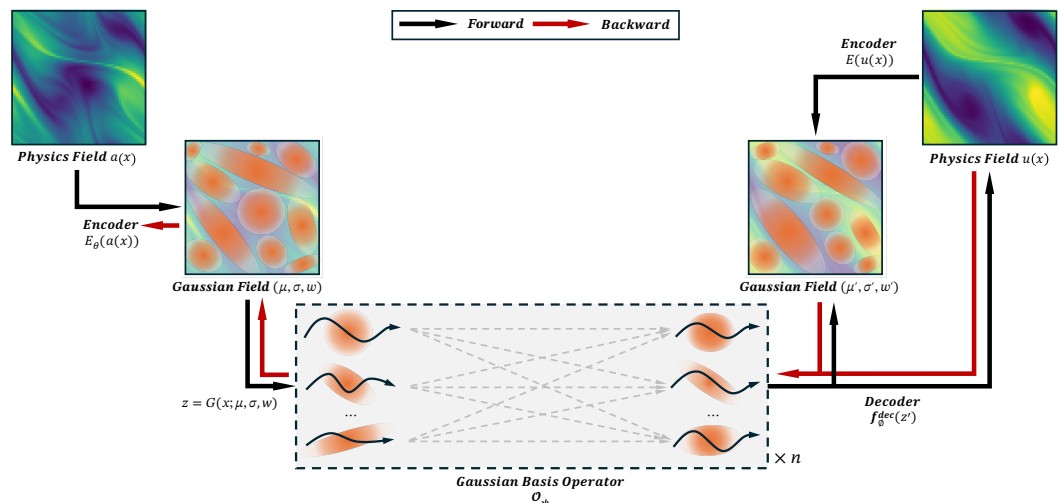

Figure 2: **Architecture of the Gaussian Particle Operator (GPO).** The pipeline encodes $a(\mathbf{x})$ into a Gaussian field $(\mu, \sigma, w)$, evaluates a basis $Z$, applies the modal operator $\mathcal{O}_\psi$, and decodes to $\hat{u}(\mathbf{x})$; black arrows denote forward computation and red arrows denote gradients.

**Expressivity of the modal operator.** We formalize that PG Gaussian Attention can approximate a broad class of continuous operators:

**Theorem 3.2** (Universal approximation in modal form). *Let $\mathcal{T} : L^p(\Omega; \mathbb{R}^{c_{in}}) \to L^q(\Omega; \mathbb{R}^{c_{out}})$ be continuous on bounded sets and admit either a Mercer/Hilbert–Schmidt kernel $K(\mathbf{x}, \mathbf{x}')$ or a low-rank factorization $\mathcal{T} \approx \Phi(\cdot)\,\mathcal{K}\,\Phi(\cdot)^\top$ with continuous $\Phi : \Omega \to \mathbb{R}^m$. Then, for any $\varepsilon > 0$, there exist a modal budget $G$ and parameters $\Theta$ of our encoder, Gaussian modal windows, PG Gaussian Attention, and decoder such that $\|\mathcal{G}_\Theta - \mathcal{T}\| < \varepsilon$ (operator norm on bounded subsets). (Proof in Appx.A.2)*

### 3.3 GAUSSIAN PARTICLE OPERATOR: OVERALL FRAMEWORK

#### 3.3.1 NEURAL OPERATOR FORMULATION

Let $\Omega \subset \mathbb{R}^d$ be the domain, $a : \Omega \to \mathbb{R}^{c_{in}}$ the input field, and $u : \Omega \to \mathbb{R}^{c_{out}}$ the target field. We model the map $a \mapsto u$ by a neural operator

$$\mathcal{G}_\Theta = f_\phi^{dec} \circ \left(\mathcal{O}_\psi\right)^{\circ n} \circ \mathcal{Z}\big(\,\cdot\,;\Pi_\theta(\cdot)\big) \circ E_\theta, \qquad \Theta = (\theta, \psi, \phi), \tag{23}$$

where:

- $E_\theta$ (encoder) extracts *Gaussian particles* $\Pi_\theta(a) = \big(\mu_\theta, \sigma_\theta, w_\theta\big)$ at queried locations;

- $\mathcal{Z}(\cdot; \Pi)$ evaluates the *Gaussian basis* and returns per-location, $G$-dimensional coefficients $Z \in \mathbb{R}^{N \times G}$ with

$$z_{j,i} = w_{j,i} \exp\Big(-\tfrac{1}{2}\big\|(\mathbf{x}_j - \mu_{j,i})/\sigma_{j,i}\big\|_2^2\Big); \tag{24}$$

- $\mathcal{O}_\psi$ is the *Gaussian-basis operator* (Sec. 3.2) acting on $Z$ and parameterized by PG Gaussian Attention; it can be applied $n$ times:

$$Z^{(0)} = Z, \qquad Z^{(k+1)} = \mathcal{O}_\psi\big(Z^{(k)}; \Pi_\theta(a)\big), \quad k = 0, \ldots, n-1; \tag{25}$$

- $f_\phi^{dec}$ (decoder) maps the updated basis to the output field values: $\hat{u}(\mathbf{x}_j) = f_\phi^{dec}\big(Z_{j,:}^{(n)}\big)$.

The construction is resolution-agnostic: for any query set $\{\mathbf{x}_j\}$—on 2D/3D grids or irregular meshes—one simply recomputes Eq.(24) and reuses the same $\mathcal{O}_\psi$ and $f_\phi^{dec}$.

Table 1: **Performance comparison with baselines on benchmarks.** $L_2$ loss is recorded.

| MODEL | NS2D | NS3D | ERA5-TEMP | ERA5-WIND U | CARRA-V10 | CARRA-SP |
|---|---|---|---|---|---|---|
| FNO | 3.24E-02 | 5.07E-01 | 7.09E-03 | 1.02E-01 | 3.50E-01 | 1.61E-03 |
| LSM | **3.11E-02** | **3.80E-01** | / | / | / | / |
| GALERKIN TRANSFORMER | 8.81E-02 | 5.39E-01 | 5.44E-03 | 1.55E-01 | 3.73E-01 | **1.34E-03** |
| GNOT | 7.19E-01 | 1.01E+00 | 1.55E-02 | 3.49E-01 | 7.57E-01 | 4.89E-03 |
| TRANSOLVER | 3.76E-02 | 5.29E-01 | 4.18E-03 | 1.06E-01 | 3.76E-01 | 1.48E-03 |
| ONO | 4.26E-02 | 8.83E-01 | 1.45E-02 | 3.49E-01 | 7.25E-01 | 5.75E-03 |
| **GPO** | 3.90E-02 | 4.21E-01 | **2.30E-03** | **6.74E-02** | **2.97E-01** | 2.15E-03 |

### 3.3.2 PIPELINE OVERVIEW

As shown in Figure 2, given an input field $a(\mathbf{x})$, the encoder $E_\theta$ produces per-site Gaussian particles $\Pi_\theta(a) = (\mu, \sigma, w)$, i.e., a *Gaussian field*. At query locations $\{\mathbf{x}_j\}_{j=1}^N$, we then evaluate the Gaussian basis by Eq.(24) to obtain $Z \in \mathbb{R}^{N \times G}$. The Gaussian-basis operator $\mathcal{O}_\psi$ acts on $Z$ in modal space and can be applied for $n$ stages as in Eq.(25) to capture multi-step coupling, yielding $Z^{(n)}$. Finally, the decoder $f_\phi^{\text{dec}}$ maps $Z_{j,:}^{(n)}$ to $\hat{u}(\mathbf{x}_j)$. During training, the target $u(\mathbf{x})$ may also be encoded by $E_\theta$ to provide an auxiliary Gaussian-field supervision signal.

## 4 EXPERIMENTS

**Benchmarks.** We evaluate on two synthetic Navier–Stokes surrogates and two real reanalyses to span 2D→3D and regular→irregular domains. **NS2D** (Kovachki et al., 2023) is an incompressible periodic box sampled on $64 \times 64$; **NS3D** (Takamoto et al., 2022) extends to a periodic cube on $64^3$, stressing 3D scalability. **ERA5** (Hersbach et al., 2023) uses one month on the $0.25°$ global grid ($721 \times 1440$), with variables 2 m temperature ($t$) and 10 m zonal wind ($u$). **CARRA** (Schyberg et al., 2020) uses one Arctic month on its native regional grid ($989 \times 789$) with an irregular land/sea/ice mask, variables 10 m meridional wind ($v_{10}$) and surface pressure ($sp$). We train one-step operators and assess multi-step rollouts, using native grids, latitude weighting for ERA5 and CARRA.

**Baselines.** We compare with two physics-inspired neural operators—**FNO** (Fourier neural operator) (Li et al., 2021) and **LSM** (Wu et al., 2023) (learned spectral mixing)—and four Transformer-based operators—**Galerkin Transformer** (Cao, 2021), **GNOT** (Hao et al., 2023), **ONO** (Xiao et al., 2024), and **Transolver** (Wu et al., 2024). The first group represents spectral/kernelized designs without attention; the second group parameterizes the operator via attention (global coupling). All baselines use identical data splits, losses, and rollout protocols, and we keep model capacity comparable; 2D/3D/Irregular variants are used where applicable.

**Implementations.** We evaluate all models using the relative $L_2$ error on held-out sets. Inputs/targets are *normalized per variable* using training statistics; models are trained on the normalized data, and *all metrics are computed after inverse normalization*. Training uses AdamW (Loshchilov & Hutter, 2019) with an initial learning rate of $10^{-3}$ and a StepLR scheduler that reduces the learning rate at fixed intervals. All experiments are run on a single NVIDIA RTX 4090 GPU. Per-dataset and per-baseline hyperparameters are provided in Appx.B.

### 4.1 BENCHMARK PERFORMANCE

Table 1 summarizes $L_2$ errors across synthetic Navier–Stokes and real reanalyses. The results align with the inductive biases of each family while highlighting the competitiveness of GPO.

### KEY OBSERVATIONS

(i) **Synthetic, regular grids.** On NS2D/NS3D, physics-inspired spectral operators excel: LSM attains the best errors on NS2D ($3.11 \times 10^{-2}$) and NS3D ($3.80 \times 10^{-1}$), with FNO close behind. *GPO is competitive* (NS2D: $3.90 \times 10^{-2}$; NS3D: $4.21 \times 10^{-1}$, second-best), despite not using fixed

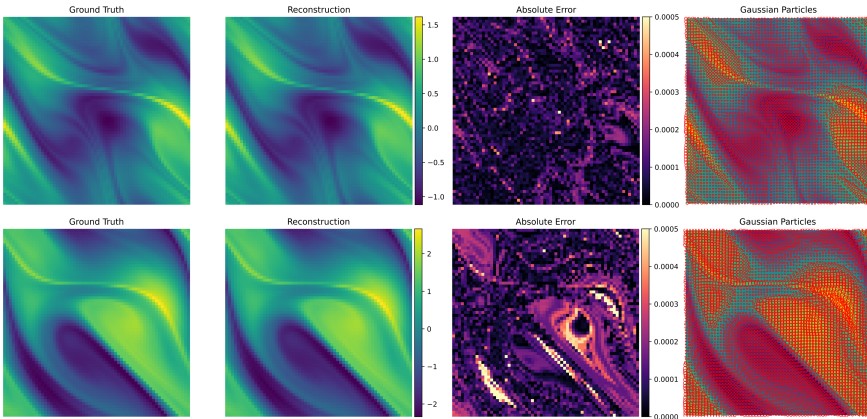

Figure 3: Interpretable visualization on an **in-distribution sample (above)** and an **out-of-distribution sample (below)**. Left to right: ground truth, reconstruction from the Gaussian basis, absolute error, and learned Gaussian particles overlaid (ellipses: center $\mu$, axes $\propto \sigma$, color/size $\propto w$).

Fourier priors—evidence that the Gaussian basis plus PG-attention can match spectral baselines on clean, regular settings. Note that, LSM achieves its gains with substantially larger capacity and cost (Table 4), indicating that GPO attains comparable accuracy with $\sim 12\times$ fewer parameters.

(ii) **Large, complex geophysical data.** On ERA5 and CARRA, *GPO and attention-based* operators outperform spectral baselines: GPO achieves the best results on *ERA5-temp* ($2.30\times10^{-3}$), *ERA5-wind u* ($6.74\times10^{-2}$), and *CARRA-v10* ($2.97\times10^{-1}$); for *CARRA-sp*, Galerkin/Transolver lead (GPO remains competitive at $2.15\times10^{-3}$). Notably, LSM cannot be applied on these grids (slashes in Table), underscoring the advantage of mesh-/mask-agnostic designs for irregular domains and spherical weighting.

The pattern is consistent with our design: the *Gaussian particle representation* provides localized, interpretable support adaptable to irregular masks and 2D→3D settings, while *PG Gaussian Attention* supplies global modal coupling without quadratic growth in spatial samples. Consequently, GPO is *on par with* spectral methods on regular synthetic benchmarks and *strongly superior* on real, large-scale datasets where irregularity and global coupling are critical.

## 4.2 INTERPRETABLE VISUALIZATION

### 4.2.1 RECONSTRUCTION

**Setup.** Before assessing the operator itself, we first verify that the learned *Gaussian (particle) basis* is both *faithful* and *interpretable*. We train the encoder–decoder (Sec. 3.1) to reconstruct the field, using the weighted Gaussian evaluation and the particle regularizers (center alignment and scale-range barrier). We then visualize, for **in-distribution (ID)** and **out-of-distribution (OOD)** cases, the ground-truth field, the reconstruction, the absolute error, and the learned particles overlaid on the field (ellipses: center = $\mu$, axes $\propto \sigma$, color/size $\propto w$).

**Observations.** As shown in Figure 3: (i) the particles concentrate along coherent structures (fronts, filaments, vortices) with *anisotropic* scales aligned to local flow directions, yielding compact yet accurate reconstructions. (ii) Error maps are predominantly localized near sharp gradients or subgrid filaments. (iii) Under OOD shifts, the representation remains stable: particle geometry and weights adapt to novel patterns, and the measured errors increase modestly while preserving large- and meso-scale features. These results substantiate our design choice of using Gaussian particles as the primary state: the basis is *visualizable, interpretable, and mesh-agnostic*, and it provides a robust trial space onto performs PG-style modal coupling.

### 4.2.2 LAYER-WISE DYNAMICS OF THE GAUSSIAN PARTICLE FIELD

**Setup.** During prediction we apply the Gaussian Particle Operator (Sec. 3.2) for $n$ stages. At stage $k$, the encoder-fixed particles $(\mu, \sigma)$ define the *trial* atoms, while the PG Gaussian Attention updates

the per-site coefficients $\mathbf{z}_j^{(k)} \in \mathbb{R}^G$ (basis activations). We visualize the *particle field* at each stage by overlaying the particle footprints (ellipses: center $= \mu$, axes $\propto \sigma$) with color proportional to the local activation (e.g., $A^{(k)}(\mathbf{x}_j) = \sum_{g=1}^{G} z_{j,g}^{(k)}$). This directly exposes how the operator *re-weights* and *redistributes* modal energy across the learned particles.

**Interpretability.** As shown in Figure 4, layer-wise changes in the activation maps therefore admit a physical reading: (i) smoothing of highly symmetric couplings resembles diffusion among nearby modes; (ii) directed transfers captured by off-diagonal attention act like advection of features along the dominant flow directions encoded by $(\mu, \sigma)$; and (iii) growth/decay of localized activations reveals cross-scale energy exchange. The bottom panels (input $a(\mathbf{x})$ vs. target $u(\mathbf{x})$) provide the macroscopic reference: the progressive adjustment of particle activations from Layer 1→Layer $n$ tracks the formation/transport of fronts and filaments that distinguish $u$ from $a$. See Appx.D for more visualization.

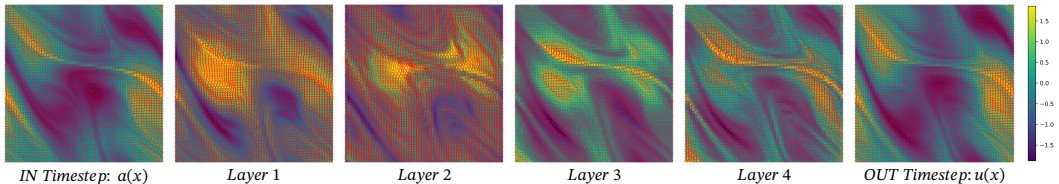

IN Timestep: $a(x)$     Layer 1     Layer 2     Layer 3     Layer 4     OUT Timestep: $u(x)$

Figure 4: **Layer-wise evolution of the Gaussian particle field.** Particle activations after successive PG Gaussian Attention layers (Layer 1→Layer 4).

### 4.3 MODEL ANALYSIS

**Ablations** (Table 3) confirm that the *Gaussian Field* is essential, while even a single Gaussian per site improves over MLP baselines. The best results arise from combining the Gaussian Field with *PG Gaussian Attention*. Increasing the modal budget $G$ generally helps but exhibits diminishing returns; we therefore adopt a moderate $G$ for a balanced accuracy–efficiency trade-off.

**Complexity** (Table 4) shows that GPO maintains low memory use and competitive runtime, scaling near-linearly with the number of query points (unlike spatial self-attention), and offering favorable trade-offs across 2D/3D and irregular domains. See Appx.C for full ablations and analyses.

## 5 CONCLUSION

We introduced the *Gaussian Particle Operator* (GPO): a resolution-agnostic neural operator that represents fields by an interpretable *Gaussian (particle) basis* and performs basis-to-basis coupling via *Petrov–Galerkin Gaussian Attention*. The design makes every intermediate object—particles $(\mu, \sigma, w)$, modal windows, and inter-modal couplings—directly visualizable, while achieving competitive (often superior) accuracy on synthetic NS2D/NS3D and large real-world datasets (ERA5, CARRA), with improved spectral fidelity and stable rollouts.

### LIMITATIONS AND FUTURE WORK

**Effectiveness & efficiency.** Performance depends on the modal budget $G$ and head width; although $N \leftrightarrow G$ transfers are linear in $N$, the memory and compute of $NG$ windows can still be a bottleneck at extreme resolutions. Future work includes adaptive or hierarchical particles (multi-scale $G$), sparse/modal pruning and routing, structured/low-rank attention in $G$-space, and optimized implementations (mixed precision, kernel fusion) to further improve accuracy–efficiency trade-offs.

**Physics integration for deeper interpretability.** The current training is primarily data-driven with lightweight particle regularization; it does not *guarantee* invariants (e.g., mass/energy) or constraints (e.g., divergence-free flow, boundary conditions). We plan to couple the Gaussian basis more tightly with physics to align particles with physically evolving structures. These aim to produce models that are not only accurate but also *mechanistically* interpretable.

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

## A   EXPRESSIVITY OF THE GAUSSIAN FIELD AND GPO

### A.1   EXPRESSIVITY OF THE GAUSSIAN FIELD

**Lemma A.1** (Density of Gaussian mixtures). *On compact $\Omega$, finite mixtures of anisotropic Gaussians are dense in $C(\Omega)$ (and dense in $L^r(\Omega)$ for $1 \le r < \infty$). Hence for any continuous scalar field $v$ and $\varepsilon > 0$, there exist $G, \{\mu_i, \sigma_i, w_i\}_{i=1}^G$ such that $\|v(\cdot) - \sum_{i=1}^G w_i \exp(-\frac{1}{2}\|(\cdot - \mu_i)/\sigma_i\|^2)\|_\infty < \varepsilon$.*

*Sketch.* Standard universal approximation results for radial basis functions/Gaussian kernels.

*Proof.* We give a constructive proof based on Gaussian mollification and Riemann sums.

**Step 1: Approximate identity via Gaussian mollifiers.**   Let $\Omega \subset \mathbb{R}^d$ be compact and let $v \in C(\Omega)$. By Tietze's extension theorem there exists $\tilde{v} \in C_c(\mathbb{R}^d)$ such that $\tilde{v}|_\Omega = v$. For $\Sigma \in \mathbb{R}^{d \times d}$ symmetric positive definite, set the (unnormalized) Gaussian

$$\phi_\Sigma(x) = \exp\left(-\tfrac{1}{2}\, x^\top \Sigma^{-1} x\right).$$

Let $\{\Sigma_\epsilon\}_{\epsilon \downarrow 0}$ be any family with $\|\Sigma_\epsilon\| \to 0$. Since Gaussians form an approximate identity, the *normalized* mollification $\tilde{v} * \phi_{\Sigma_\epsilon} / \int_{\mathbb{R}^d} \phi_{\Sigma_\epsilon}$ converges to $\tilde{v}$ *uniformly* on compact sets as $\epsilon \downarrow 0$ (uniform continuity of $\tilde{v}$ and standard approximate-identity properties). Because the normalization constant is a positive scalar depending only on $\Sigma_\epsilon$, we can absorb it into the mixture weights later. Hence, for any $\eta > 0$ there exists $\epsilon_0$ such that for all $0 < \epsilon \le \epsilon_0$,

$$\sup_{x \in \Omega} \left| (\tilde{v} * \phi_{\Sigma_\epsilon})(x) - \tilde{v}(x) \right| < \tfrac{\eta}{2}. \tag{26}$$

**Step 2: Riemann-sum approximation of the convolution (finite mixture).**   Fix such an $\epsilon$, write $\Sigma = \Sigma_\epsilon$, and denote the convolution

$$(\tilde{v} * \phi_\Sigma)(x) = \int_{\mathbb{R}^d} \tilde{v}(y)\, \phi_\Sigma(x - y)\, dy.$$

Since $\tilde{v}$ is compactly supported and continuous while $\phi_\Sigma$ is continuous and rapidly decaying, the integrand is continuous with compact support in $y$ uniformly in $x \in \Omega$. Hence Riemann sums approximate the integral uniformly in $x$: there exists a finite set of nodes $\{\mu_i\}_{i=1}^G \subset \mathbb{R}^d$ with associated positive quadrature weights $\{\Delta_i\}_{i=1}^G$ such that

$$\sup_{x \in \Omega} \left| (\tilde{v} * \phi_\Sigma)(x) - \sum_{i=1}^G \tilde{v}(\mu_i)\, \phi_\Sigma(x - \mu_i)\, \Delta_i \right| < \tfrac{\eta}{2}. \tag{27}$$

Define mixture weights $w_i := \tilde{v}(\mu_i)\, \Delta_i$ (real-valued; the lemma does not restrict their sign), and note that each term is exactly a (shared-covariance) Gaussian atom $\exp\left(-\frac{1}{2}\|(x - \mu_i)\|_{\Sigma^{-1}}^2\right)$, i.e.,

$$\sum_{i=1}^G w_i\, \phi_\Sigma(x - \mu_i) = \sum_{i=1}^G w_i \exp\left(-\tfrac{1}{2}(x - \mu_i)^\top \Sigma^{-1}(x - \mu_i)\right).$$

**Step 3: Uniform approximation on $\Omega$.**   Combining equation 26 and equation 27,

$$\sup_{x \in \Omega} \left| \tilde{v}(x) - \sum_{i=1}^G w_i \exp\left(-\tfrac{1}{2}(x - \mu_i)^\top \Sigma^{-1}(x - \mu_i)\right) \right|$$

$$\le \sup_{x \in \Omega} \left| \tilde{v}(x) - (\tilde{v} * \phi_\Sigma)(x) \right| + \sup_{x \in \Omega} \left| (\tilde{v} * \phi_\Sigma)(x) - \sum_i w_i \phi_\Sigma(x - \mu_i) \right|$$

$$< \eta.$$

Restricting back to $\Omega$ (where $\tilde{v} = v$) yields

$$\left\| v(\cdot) - \sum_{i=1}^G w_i \exp\left(-\tfrac{1}{2}\|(\cdot - \mu_i)\|_{\Sigma^{-1}}^2\right) \right\|_\infty < \eta.$$

Since $\eta > 0$ was arbitrary, finite mixtures of (possibly anisotropic) Gaussians are dense in $C(\Omega)$.

**Anisotropy and vector-valued extension.** We used a common covariance $\Sigma$ for clarity; allowing mode-dependent $\Sigma_i$ only increases expressivity, so the same result holds with per-atom anisotropy. For vector-valued $v$, apply the scalar result componentwise.

$L^r$ **density.** Because $C(\Omega)$ is dense in $L^r(\Omega)$ for $1 \le r < \infty$ on compact $\Omega$, the uniform approximation implies $L^r$ approximation, completing the proof. $\qquad\square$

## A.2 EXPRESSIVITY OF GPO

**Theorem A.2** (Universal approximation in modal form). *Let $\mathcal{T} : \mathcal{X} \to \mathcal{Y}$ be a continuous operator on compacta that admits a Hilbert–Schmidt (Mercer-type) kernel $K(\mathbf{x}, \mathbf{x}')$ or, more generally, a low-rank factorization $\mathcal{T} \approx \Phi(\cdot)\,\mathcal{K}\,\Phi(\cdot)^\top$ with continuous features $\Phi : \Omega \to \mathbb{R}^m$. Then, for any $\varepsilon > 0$, there exist $G$ and network parameters $\Theta$ such that $\|\mathcal{G}_\Theta - \mathcal{T}\|_{\mathcal{X}\to\mathcal{Y}} < \varepsilon$.*

*Sketch.* By Lemma A.1 and universal approximation of MLPs, windows $p(\mathbf{x}, g)$ and latent features $S(Z(\mathbf{x}))$ approximate $\Phi(\mathbf{x})$; attention realizes a trainable $\mathcal{K}$ on the $G$ modes. The scatter-and-decoder emulate the output feature map. Increasing $G$ and widths yields density in the space of continuous operators.

*Proof.* We prove the claim for operators on compact domains by reducing to a finite–rank Mercer approximation and showing that each stage of our pipeline can approximate the corresponding finite–dimensional objects arbitrarily well. Throughout, $\|\cdot\|_{\mathcal{X}\to\mathcal{Y}}$ denotes the operator norm on bounded subsets.

**Step 0: Mercer (or low–rank) truncation.** Assume $\mathcal{T}$ is continuous on bounded sets and admits either a Hilbert–Schmidt kernel $K(\mathbf{x}, \mathbf{x}')$ or, more generally, a low–rank factorization $\mathcal{T} \approx \Phi(\cdot)\,\mathcal{K}\,\Phi(\cdot)^\top$ with continuous $\Phi : \Omega \to \mathbb{R}^m$. In the Mercer case, by spectral theory,

$$K(\mathbf{x}, \mathbf{x}') = \sum_{r=1}^{\infty} \lambda_r\, \varphi_r(\mathbf{x})\, \varphi_r(\mathbf{x}'), \quad \lambda_r \ge 0, \ \{\varphi_r\} \subset C(\Omega),$$

and the partial sums define finite–rank operators $\mathcal{T}_m f(\mathbf{x}) = \sum_{r=1}^{m} \lambda_r \varphi_r(\mathbf{x}) \int \varphi_r(\mathbf{x}') f(\mathbf{x}')\, d\mathbf{x}'$ with $\|\mathcal{T} - \mathcal{T}_m\| \to 0$ as $m \to \infty$ (uniform on compacta). In the given low–rank form, select $m$ and continuous $\Phi_m : \Omega \to \mathbb{R}^m, \mathcal{K}_m \in \mathbb{R}^{m \times m}$ such that

$$\left\|\mathcal{T} - \mathcal{T}_m\right\| < \varepsilon/3, \qquad \mathcal{T}_m f(\mathbf{x}) = \Phi_m(\mathbf{x})\,\mathcal{K}_m \int_\Omega \Phi_m(\mathbf{x}')^\top f(\mathbf{x}')\, d\mathbf{x}'. \tag{28}$$

**Step 1: Approximating the feature maps by Gaussian basis + MLPs.** By Lemma A.1 (density of Gaussian mixtures) and universal approximation of MLPs, for any $\delta > 0$ there exist: (i) a point-wise encoder/evaluator producing $Z(\mathbf{x}) \in \mathbb{R}^G$ from Gaussian particles $(\mu, \sigma, w)$ and a small MLP $S$ such that the *trial features* $\Psi(\mathbf{x}) \in \mathbb{R}^D$, defined by $\Psi(\mathbf{x}) = S(Z(\mathbf{x}))$, satisfy

$$\sup_{\mathbf{x}\in\Omega} \left\|\Psi(\mathbf{x}) - \Phi_m(\mathbf{x})\right\|_2 < \delta; \tag{29}$$

(ii) head–wise *Gaussian modal windows* $p(\mathbf{x}, g) \ge 0$ with $\sum_{g=1}^{G} p(\mathbf{x}, g) = 1$, implemented by linear maps on $[Z(\mathbf{x}), (\mu, \sigma, w)(\mathbf{x})]$ and a softmax, such that the *test* functionals

$$\mathcal{M}_g(f) = \int_\Omega p(\mathbf{x}, g)\, f(\mathbf{x})\, d\mathbf{x}$$

approximate the $m$ target coordinates $\int \Phi_m(\mathbf{x})^\top f(\mathbf{x})\, d\mathbf{x}$ after a fixed linear readout. Concretely, there exists $W \in \mathbb{R}^{m \times G}$ with $\| W [\mathcal{M}_g(\cdot)]_{g=1}^G - \int \Phi_m(\cdot)^\top(\cdot) \| < C_1 \delta$. (One can view $W p(\mathbf{x}, \cdot)$ as a learned quadrature/test family for the $m$ coordinates.)

**Step 2: Discrete PG measurement and quadrature error.** Given a discretization $\{\mathbf{x}_j\}_{j=1}^N$ with empirical measure converging to the sampling measure on $\Omega$, the $N \to G$ aggregation used in Sec. 3.2 forms tokens

$$t_g = \frac{\sum_{j=1}^N p(\mathbf{x}_j, g)\, \Psi(\mathbf{x}_j)}{\sum_{j=1}^N p(\mathbf{x}_j, g)} \quad \in \mathbb{R}^D.$$

By uniform continuity of $\Psi$ and $p(\cdot, g)$ on compact $\Omega$, Riemann (or Monte Carlo) sums converge to the integrals. Hence there exists $N_0$ so that for all $N \geq N_0$,

$$\left\| \left[ t_g \right]_{g=1}^{G} - \left[ \frac{\int p(\mathbf{x}, g)\, \Psi(\mathbf{x})\, d\mathbf{x}}{\int p(\mathbf{x}, g)\, d\mathbf{x}} \right]_{g=1}^{G} \right\| < C_2 \delta. \tag{30}$$

Post-multiplying by $W$ and using equation 29 shows that the vector of $m$ measured coordinates is within $C_3 \delta$ of $\int \Phi_m(\mathbf{x})^\top f(\mathbf{x})\, d\mathbf{x}$ for any $f$ in a bounded set.

**Step 3: Implementing the modal coupling by attention + linear maps.** We next show that the $G \times G$ *modal attention* stage can realize the finite linear map $\mathcal{K}_m$ (up to basis changes) to arbitrary precision. Using the head projections $W_z$ and $W_{\text{out}}$, the attention block computes

$$\widetilde{T} = \alpha \left( T W_V \right), \qquad Y = \left( \widetilde{T} \right) W_{\text{out}},$$

where $T \in \mathbb{R}^{G \times D}$ stacks the tokens $t_g$, $\alpha$ is the softmax attention matrix, and $W_V, W_{\text{out}}$ are learned linear maps. Since softmax can approximate a Kronecker–delta (by sending on–diagonal logits to $+\infty$ and off–diagonal to $-\infty$), we can set $\alpha \approx I_G$ arbitrarily closely. Then $Y \approx T(W_V W_{\text{out}})$. Because $W_V, W_{\text{out}}$ are unconstrained, their product can approximate any target matrix $M \in \mathbb{R}^{D \times m}$ to arbitrary precision. Choosing $M$ to implement the composition $W \mathcal{K}_m$ (after the measurement map from Step 2), we obtain a block that emulates $v \mapsto \mathcal{K}_m v$ in the $m$–dimensional modal coordinates. (If desired, one may keep $\alpha$ nontrivial and absorb its effect into the surrounding linear maps; the argument is unchanged.)

**Step 4: Scatter and pointwise decoding.** The $G \to N$ scatter re-distributes the mixed modal features back to locations via the same windows $p(\mathbf{x}, g)$, followed by a pointwise decoder MLP $f_\phi^{\text{dec}} : \mathbb{R}^G \to \mathbb{R}^{c_{\text{out}}}$. Since MLPs are universal approximators on compacta, the composition can approximate the desired output feature map $\mathbf{x} \mapsto \Phi_m(\mathbf{x})$ (or its linear image) uniformly, matching the form in equation 28.

**Step 5: Error aggregation.** Let $\varepsilon_m = \|\mathcal{T} - \mathcal{T}_m\| < \varepsilon/3$ be the truncation error. Pick $\delta > 0$ sufficiently small and $N$ sufficiently large so that: (i) the feature/window approximations introduce at most $C\delta$ error in the measured coordinates (Steps 1–2), (ii) the attention+linear block approximates the modal coupling $\mathcal{K}_m$ within $C\delta$ uniformly on bounded sets (Step 3), and (iii) the scatter+decoder approximates the output features within $C\delta$ uniformly (Step 4). By stability (continuity) of all stages,

$$\|\mathcal{G}_\Theta - \mathcal{T}\| \leq \underbrace{\|\mathcal{G}_\Theta - \mathcal{T}_m\|}_{\leq C\delta} + \underbrace{\|\mathcal{T}_m - \mathcal{T}\|}_{\varepsilon_m} < C\delta + \varepsilon/3.$$

Choosing $\delta$ so that $C\delta < 2\varepsilon/3$ yields $\|\mathcal{G}_\Theta - \mathcal{T}\| < \varepsilon$.

Combining the steps completes the proof. $\square$

# B  Implementation Details

## B.1  Baseline implementations

All baseline models (FNO, LSM, Galerkin Transformer, GNOT, ONO, Transolver) are adapted from the *Neural-Solver-Library* (Wu et al., 2024) reference implementation at `https://github.com/thuml/Neural-Solver-Library`. Unless otherwise noted, we keep an identical training schedule across baselines: AdamW optimizer, initial learning rate $10^{-3}$ with a `StepLR` scheduler (step size and decay factor as in the library's default per dataset), up to 500 epochs with validation early stopping, the same data normalization/inverse-normalization protocol, and matched rollout/evaluation settings.

## B.2  GPO configurations

The dataset-specific configurations of GPO are summarized in Table 2. We provide the source code of GPO in the Supplementary Material.

Table 2: **Model configurations of GPO.**

| BENCHMARKS | MODEL CONFIGURATIONS | | | |
|---|---|---|---|---|
| | HIDDEN_DIM | NUM_LAYERS | NUM_HEADS | NUM_GAUSSIANS |
| NS2D | 128 | 8 | 8 | 32 |
| NS3D | 64 | 8 | 4 | 16 |
| ERA5-TEMP | 64 | 4 | 4 | 16 |
| ERA5-WIND U | 64 | 4 | 4 | 16 |
| CARRA-V10 | 64 | 4 | 4 | 16 |
| CARRA-SP | 64 | 4 | 4 | 16 |

## C MODEL ANALYSIS

### C.1 ABLATION STUDY

Table 3 reports the $L_2$ error under controlled variants (parameter counts are adjusted to be comparable). **(i) Necessity of the Gaussian Field.** Replacing the Gaussian Field with plain MLP encoder/decoder (`w/o Gaussian Field`) degrades accuracy markedly ($7.44 \times 10^{-2}$), and removing the PG operator while keeping the Gaussian Field (`w/o PG Operator`) is even worse ($8.57 \times 10^{-2}$). Notably, even a *single* Gaussian per site (`num_gaussian=1`) already improves to $6.28 \times 10^{-2}$, indicating that particleized Gaussian evaluation is a beneficial inductive bias beyond a black-box MLP. **(ii) Synergy of PG Operator and Gaussian Field.** Combining the Gaussian basis with the PG Gaussian Attention yields the full GPO (baseline: $3.90 \times 10^{-2}$), demonstrating that the PG measurement→modal coupling→scatter complements the local particle representation; each component alone is insufficient. **(iii) Effect of the number of Gaussians.** Increasing `num_gaussian` consistently reduces error (from $4.21 \times 10^{-2}$ at $G=16$ to $3.84 \times 10^{-2}$ at $G=64$), but with diminishing returns; considering cost (Sec. C.2), we adopt $G=16/32$ as a practical trade-off between efficiency and accuracy.

Table 3: Ablation results comparing the $L_2$ error of different configurations.

| MODEL CONFIGURATION | $L_2$ ERROR |
|---|---|
| W/O PG OPERATOR | 8.57E-02 |
| W/O GAUSSIAN FIELD | 7.44E-02 |
| NUM_GAUSSIAN = 1 | 6.28E-02 |
| NUM_GAUSSIAN = 16 | 4.21E-02 |
| NUM_GAUSSIAN = 64 | 3.84E-02 |
| GPO (BASELINE) | 3.90E-02 |

### C.2 COMPUTATIONAL COMPLEXITY

Empirical measurements (Table 4, $64 \times 64 \times 3$, batch 16) corroborate the analysis: GPO attains low memory footprint (2,313 MiB) and competitive time (44.66 s/epoch train; 1.67 s/epoch inference) with a modest parameter count (6.10 MB), outperforming attention baselines in training speed (Galerkin/Transolver/ONO/GNOT) and GPU memory, while remaining close to spectral baselines at inference. Although FNO is fastest on this small grid, GPO's cost grows near–linearly with $N$ and remains stable when moving to higher resolutions or 3D, where spatial attention becomes prohibitive and FFT memory/IO costs rise.

By aggregating *locally* ($N \leftrightarrow G$) and coupling *globally* only in modal space ($G \times G$), GPO delivers resolution–agnostic efficiency: linear scaling in $N$, controllable quadratic dependence on $G$, and favorable memory/time trade–offs across 2D/3D and irregular domains.

Table 4: **Computational efficiency comparison across models** (measured with input size 64×64×3, batch size 16).

| MODEL | PARAM COUNT | PARAM (MB) | GPU MEM (MiB) | TRAIN (S/EPOCH) | INFERENCE (S/EPOCH) |
|---|---|---|---|---|---|
| FNO | 640,305 | 4.84 | 949 | 28.27 | 0.5 |
| LSM | 19,187,457 | 73.23 | 2,875 | 48.42 | 1.73 |
| GALERKIN TRANSFORMER | 1,096,321 | 4.18 | 4,301 | 65.29 | 2.96 |
| GNOT | 2,485,901 | 9.48 | 8,643 | 139.42 | 6.09 |
| TRANSOLVER | 3,069,889 | 11.71 | 4,917 | 97.03 | 4.10 |
| ONO | 1,596,673 | 6.09 | 6,163 | 94.80 | 4.27 |
| **GPO (OURS)** | 1,598,257 | 6.10 | 2,313 | 44.66 | 1.67 |

# D ADDITIONAL VISUALIZATIONS

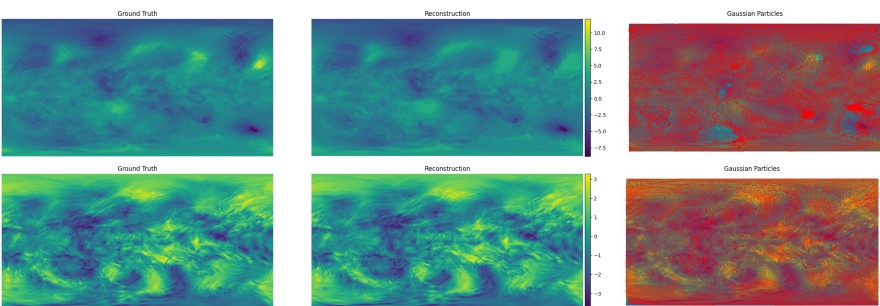

Figure 5: Interpretable visualization of ERA5 on an **in-distribution sample (above)** and an **out-of-distribution sample (below)**. Left to right: ground truth, reconstruction from the Gaussian basis and learned Gaussian particles overlaid (ellipses: center $\mu$, axes $\propto \sigma$, color/size $\propto w$).

