# OpenReview forum: "From Basis to Basis: Gaussian Particle Representation for Interpretable PDE Operators"
_ICLR.cc/2026/Conference — Submitted to ICLR 2026_

### Official Review · Reviewer_whp9 · 2025-10-22

**Soundness:** 2
**Presentation:** 1
**Contribution:** 2
**Rating:** 2
**Confidence:** 3

**Summary:**

Radial basis (with learnable diagonal covariance) representation followed by an Attention-based operator.

**Strengths:**

- Center alignment loss is good to impose a meaning in Gaussian particle positions.
- Evaluation on a realistic dataset, e.g., ERA5.
- Universal approximation theorem.

**Weaknesses:**

- Figure 1: Since non-diagonal covariance has not been considered, rotated Gaussians are impossible.
- Section 3.2: I can’t find any explicit connections between the PG method. The described idea is 1) computing the coefficient for the Gaussian representation, 2) applying attention several times to the coefficients, and 3) reconstruction.
- Park and Sandberg took a similar approach for Lemma 1. It is necessary to indicate what is new and what is brought from existing literature. (Also Theorem 3.2.)
- It is necessary to include and discuss [PIG: Physics-Informed Gaussians as Adaptive Parametric Mesh Representations, Kang et al., ICLR 2025] and references therein.

**Questions:**

Questions
- Line 97: How has frequency bias been addressed by GPO?
- How were out-of-distribution (OOD) cases defined?
- How were $\sigma_{min}$ and $\sigma_{max}$ chosen?
- Ablation study for alignment loss?
- "GPO’s cost grows nearly linearly with N" -> could you provide a table? Can’t find in Table 4 (Line 462—463).
- Line 134: “computationally, the map ~ is local.” In my understanding, it is necessary to evaluate all Gaussians for $\tilde{x}$. Thus, I would say conceptually local, but computationally global. Please correct me if I am wrong.
- The authors mentioned interpretability and visualizability as two of the advantages of GPO. I agree with "visualizability." But I am wondering what kind of meaningful interpretation is possible, other than that from visualizability.
- Line 179: “at each location j, G weighted coefficients $z_j$ are derived from particles.” Why $z_j$ coefficients? (Line 206 also)
- Eq. 11 isn’t the tilde notation for query points?

Suggestions
- Line 108: G overlaps: Gaussian and the number of that.
- Eq. 3-6 notations can be improved.
- Line 180: The PG acronym was defined multiple times.
- Eq. 10 notation can be improved, e.g., $(x-\mu)^T \Sigma^{-1}(x-\mu)$
- Line 119: $a_j := a(x_j)$
- Eq. 20 typo?

---

### Official Review · Reviewer_btKh · 2025-10-27

**Soundness:** 3
**Presentation:** 3
**Contribution:** 2
**Rating:** 4
**Confidence:** 4

**Summary:**

This work introduces the Gaussian Particle Operator (GPO), a neural operator framework that represents PDE fields via a set of learned Gaussian particles with explicit geometric parameters The model combines this representation with a Petrov–Galerkin Attention mechanism that acts in modal space, achieving resolution-agnostic, close to linear complexity. The paper provides a detailed mathematical formulation, expressivity proofs, and experiments on several PDE benchmarks, including Navier–Stokes (2D/3D) and large-scale datasets (ERA5, CARRA).

Overall, the paper is technically sound and ambitious, aiming to bridge interpretability, scalability, and performance in operator learning. It highlights several appealing properties: explicit interpretability of latent states, mesh- and resolution-agnostic design, and theoretically motivated PG attention.

**Strengths:**

- The paper introduces a novel representation of PDE fields as learnable Gaussian mixtures. While Gaussian kernels have been used in kernel neural operators and multigrid models, this work is the first to make them the primary latent state and to combine them with a Petrov–Galerkin attention formulation.

- The connection between neural operator learning and Petrov–Galerkin projection is insightful and contributes to interpretability and theoretical grounding.

- The paper is mathematically rigorous: derivations of the Gaussian basis, PG operator formulation, and expressivity theorems are thorough and clearly presented.

- The scalability analysis is detailed (complexity reduction)

- The ablation studies convincingly isolate the contributions of the Gaussian field and PG Gaussian Attention.

- The paper is clearly written and well organized, with a logical flow from representation → operator → experiments.

- Its design is promising for large-scale geophysical or climate modeling tasks where interpretability and mesh-agnostic scalability are key.

- The visualization of Gaussian particles and modal activations provides rare interpretability in operator learning.

**Weaknesses:**

1) Missing comparisons to prior Gaussian-kernel operator works

The use of Gaussian kernels in PDE learning is not entirely novel; several recent operator frameworks rely on Gaussian or RBF kernels as integral components [1][2][3][4][5]

While M²NO and MGNO are briefly mentioned in the related-work section, no experimental or architectural comparison is made.
The authors should clarify how GPO differs (e.g., explicit particle state vs. fixed Gaussian smoothing) and whether it outperforms or scales better than these models.

2) Limited diversity of baselines

GPO is compared only with spectral (FNO, LSM) and Transformer-based operators.

State-of-the-are convolutional methods, such as the Convolutional Neural Operator [6] and Generalized Multi-Scale PDE model [7], are missing. These methods are known to perform extremely well on advection-dominated problems.

Graph-based operators (e.g., Graph Kernel NO [8], Geometry-Aware Operator Transformer [9]) are natural and scalable baselines for irregular/non-uniform domains and should be included for completeness.

3) Experimental scope

All experiments are on smooth, fluid-like datasets.

It remains unclear how GPO performs on multi-scale turbulent data (see [10] for instance), or on elliptic PDEs (e.g., Poisson, Helmholtz) that test spatial conditioning rather than temporal dynamics.

The datasets (ERA5 and CARRA) are not fully described in the text. Details such as data size, temporal horizon, and number of rollout steps are missing; they should be reported in the appendix.

While NS2D has per-epoch timing/memory (Table 4), analogous runtime/memory/horizon details for NS3D, ERA5, CARRA are not reported. Given the scalability claim, those numbers would be compelling.

The paper claims “improved spectral fidelity and stable rollouts”, but there is no explicit quantitative evaluation of rollout stability (e.g., error vs. time-step curves).

4) Performance gaps

GPO outperforms baselines only on ERA5 (and partly on CARRA) but not on NS2D/3D (Table 1).
Since the model’s main strengths are interpretability and scalability rather than accuracy, stronger baselines are needed to demonstrate clear trade-offs. It would be useful to examine how the model scales with an increasing number of training samples and larger model sizes.

5) Visualization insufficiency

Figure 3 and 5 provide only one in-distribution and one OOD visualization. More qualitative comparisons on all datasets (NS3D, ERA5, CARRA) would make interpretability claims more convincing.

____

[1] Chen, Y., & Zhang, L. (2024). Dynamic Gaussian Graph Operator (DGGO): Gaussian kernel integral operators for function-space learning

[2] Wang, Y., Liu, H., & Li, Z. (2024). Kernel Neural Operator (KNO): Learning nonlinear operators with deep kernel integral mappings

[3] Xu, R., Chen, H., & Anandkumar, A. (2024). Operator Learning with Gaussian Processes

[4] Li, Z., Lai, Z., Zhang, X., & Wang, W. (2024). M²NO: Multiresolution Operator Learning with Multiwavelet-Based Algebraic Multigrid.

[5] He, J., Liu, X., & Xu, J. (2024). MGNO: Efficient Parameterization of Linear Operators via Multigrid.

[6] Raonic, B., Molinaro, R., De Ryck, T., Rohner, T., Bartolucci, F., Alaifari, R., ... & de Bézenac, E. (2023). Convolutional neural operators for robust and accurate learning of PDEs. Advances in Neural Information Processing Systems, 36, 77187-77200.

[7] Gupta, J. K., & Brandstetter, J. (2022). Towards multi-spatiotemporal-scale generalized pde modeling. arXiv preprint arXiv:2209.15616.

[8] Li, Z., Kovachki, N., Azizzadenesheli, K., Liu, B., Bhattacharya, K., Stuart, A., & Anandkumar, A. (2020). Neural operator: Graph kernel network for partial differential equations. arXiv preprint arXiv:2003.03485.

[9] Wen, S., Kumbhat, A., Lingsch, L., Mousavi, S., Zhao, Y., Chandrashekar, P., & Mishra, S. (2025). Geometry aware operator transformer as an efficient and accurate neural surrogate for pdes on arbitrary domains. arXiv preprint arXiv:2505.18781.

[10] Herde, M., Raonic, B., Rohner, T., Käppeli, R., Molinaro, R., de Bézenac, E., & Mishra, S. (2024). Poseidon: Efficient foundation models for pdes. Advances in Neural Information Processing Systems, 37, 72525-72624.

**Questions:**

1) How does the proposed Gaussian particle representation behave when modeling fields with sharp gradients or discontinuities? Does the model sufficiently capture such features?

2) Could the authors provide quantitative evidence for the claimed rollout stability, e.g., error over multi-step horizons?

3) What is the computational footprint (GPU memory, runtime, etc) for the large-scale ERA5 and CARRA experiments compared to NS2D/NS3D?

4) Can the authors clarify the training setup for ERA5 and CARRA (dataset size, time resolution, number of rollouts at inference, and prediction horizon)?

The authors should also review the Weaknesses section for additional (implicit) questions and points raised.

**Details Of Ethics Concerns:**

/

---

### Official Review · Reviewer_7G4p · 2025-10-30

**Soundness:** 3
**Presentation:** 4
**Contribution:** 3
**Rating:** 4
**Confidence:** 4

**Summary:**

This paper proposes the Gaussian Particle Operator (GPO), a neural operator framework that represents physical fields using interpretable Gaussian particles (with centers, scales, and weights) and applies Petrov–Galerkin (PG) Gaussian Attention for efficient basis-to-basis coupling. The method aims to provide interpretability and near-linear complexity compared with conventional Transformer-based operators. Experiments on PDE benchmarks (2D/3D Navier–Stokes, ERA5, CARRA) show that GPO achieves accuracy competitive with state-of-the-art operators such as FNO, LSM, and Galerkin Transformer, while offering a more interpretable representation of flow structures.

**Strengths:**

- The use of Gaussian particles as latent bases for neural operators provides an interpretable and geometrically meaningful representation of the field.

- The design of PG Gaussian Attention links variational numerical methods and attention mechanisms, which is conceptually appealing.

- The computational analysis shows near-linear scaling with respect to the number of spatial points, which is beneficial for high-resolution or 3D applications.

**Weaknesses:**

- Marginal performance improvement: Despite the interesting design, GPO’s accuracy gains are minor. For example, on NS2D/NS3D, FNO—with fewer parameters—performs better or comparably, suggesting limited practical benefit.

- Outdated or insufficient baselines: The comparisons rely on older models (FNO 2021, LSM 2023). Recent linear-complexity Transformers such as
[arXiv:2310.01082] and [arXiv:2502.16249] (and many others) demonstrate superior efficiency and accuracy, and should be included for a fair evaluation.

- Incomplete ERA5 evaluation: Only t and u variables are reported, while ERA5 provides many other physically relevant quantities (v, z, humidity, etc.). Reporting all would give a clearer picture of generalization.

- Lack of strong empirical motivation: The claimed interpretability advantage is qualitative; there is no quantitative metric showing that the Gaussian particle representation improves diagnostic insight or prediction stability.

- Limited novelty in practice: The Gaussian-basis idea resembles radial-basis or kernel expansions, and the theoretical expressivity results repeat well-known universal-approximation arguments [arxiv.org/abs/2412.05994]. The overall innovation feels incremental.

**Questions:**

- In the proposed Petrov–Galerkin formulation, the trial and test spaces should ideally form a dual pair to ensure stability. How is this duality theoretically justified in your model? The paper defines Gaussian trial functions and Gaussian modal windows as test functions, but it is unclear whether they constitute a stable trial–test pairing. Please provide a formal argument or empirical verification demonstrating that the PG structure is indeed valid rather than only heuristic.

- The introduction highlights the issue of frequency bias in Transformer-based neural operators, yet there is no theoretical or empirical evidence showing that your Gaussian representation mitigates it. Can you provide quantitative results demonstrating improved handling of high-frequency components? For example, experiments on high–Reynolds-number Navier–Stokes equations or high-frequency Burgers/Kelvin–Helmholtz problems would directly support this claim.

- The paper frequently mentions interpretable Gaussian particles and physically meaningful modal couplings, but there is no quantitative experiment or metric assessing interpretability. How do you define physical interpretability in measurable terms? Beyond visualization, can you show that the Gaussian particle features (e.g., centers, scales, or weights) correlate with physical quantities such as vorticity or coherent structures?

---

### Official Review · Reviewer_BpAa · 2025-11-01

**Soundness:** 2
**Presentation:** 2
**Contribution:** 3
**Rating:** 4
**Confidence:** 3

**Summary:**

This paper introduces the Gaussian Particle Operator (GPO), a neural operator for learning PDE dynamics using learnable Gaussian basis functions with explicit geometric parameters (centers $\mu$, anisotropic scales $\sigma$, mixture weights $w$) that are visualizable and mesh-agnostic. GPO employs Petrov-Galerkin attention in modal space: Gaussian windows aggregate $N$ spatial locations into $G \ll N$ modal tokens, perform $G \times G$ self-attention, then scatter back to spatial locations, achieving linear complexity. The method has nice guarantees and seems to perform competitively with established methods on benchmark problems.

**Strengths:**

- The method is well-formulated, with relevant theoretical proofs (e.g., universal approximation) grounding the model design . It demonstrates competitive performance against established neural operator baselines on the tested problems .
- The interpretability of the representation is a key strength. Figures 3 and 4 nicely shows the learned particles aligning with physical structures like fronts and filaments.
- The bottleneck design of performing the compression into $G$ modal windows is a smart choice to reduce the computational cost and seems to play favorably on the front of training costs / memory cost of the model during training and inference as shown in Table 4.

**Weaknesses:**

- The selection of benchmarks could be strengthened. To fully evaluate the method's efficacy against the current state-of-the-art and ensure fair comparisons, it would be beneficial to test it on more recent, standardized benchmarks. The datasets from 'TheWell', for example, would be an excellent candidate for this.
- While the paper claims that rollout stability is a strength, these claims are not substantiated. The paper provides no details on the rollout windows evaluated. How long can the method evolve auto-regressively before it diverges due to accumulated error?
- While the paper claims that this methodology should is able to handle spectral bias that plagues neural operator methods, I believe there's more empirical results required. A convincing demonstration would include an analysis of the error spectrum, such as L2 error across different spectral buckets or a RASPD plot, to quantitatively show an advantage over baselines like FNO.
- I think the claim of tackling irregular geometries isn't well tested. As I understand it, the CARRA dataset still has a regular grid structure with masking. Would it be possible to demonstrate the performance on an unstructured mesh problem such as those found in GeoFNO?

**Questions:**

Please address the weaknesses. Additionally, I have the following questions:
- Could you perhaps discuss about the seamless capability of the model to adapt from 2D to 3D? Would this also mean that the problem should be able to adapt from information on a 2D version of the problem to a 3D one. That is NS2D can be built upon to NS3D?
- The encoder is an MLP that operates point-wise. This is counter-intuitive, as particle parameters ($\sigma$) seem to depend on neighborhood information. Did you experiment with replacing this with a neighborhood-aware encoder, such as a CNN?
- Have you studied how the model's performance scales with overall model size (total parameter count)? For example, standard Transformers exhibit well-defined scaling laws. Have you analyzed how GPO's performance scales when you simultaneously increase other parameters like model width and depth

---

### Meta-Review · Area_Chair_NnXq · 2026-01-07

**Summary:**

This paper proposes to represent PDE fields using learnable Gaussian basis functions with explicit geometric parameters. However, the performance improvements over existing methods are marginal and inconsistent across benchmarks. Besides, some key claims  about rollout stability, spectral bias mitigation, irregular geometry handling lack quantitative validation. Furthermore, the baselines are considered outdated, missing recent linear-complexity Transformers and convolutional neural operators that would provide fairer comparisons. The authors didn’t provide a rebuttal. The AC suggests rejection.

**Reviewer Concerns:**

No rebuttal is provided.

**Reviewer Scores:**

No rebuttal is provided.

---

### Decision · Program_Chairs · 2026-01-26

Reject